# Influence of the Initial Sugar Concentration and Supplementation with Yeast Extract on Succinic Acid Fermentation in a Lactose-Based Medium

Christiane Terboven [1], Christian Abendroth [2,3], Janin Laumer [1], Christiane Herrmann [4], Roland Schneider [4], Patrice Ramm [5], Joachim Venus [4] and Matthias Plöchl [1,*]

1 Bioenergie Beratung Bornim (B3), Max-Eyth-Allee 101, 14469 Potsdam, Germany; ct@b3-bornim.de (C.T.); jl@b3-bornim.de (J.L.)
2 Institute of Waste Management and Circular Economy, Technische Universität Dresden, 01796 Pirna, Germany; christian.abendroth@tu-dresden.de
3 Robert Boyle Institute e.V., Im Steinfeld 10, 07751 Jena, Germany
4 Leibniz Institute for Agricultural Engineering and Bioeconomy (ATB), Max-Eyth-Allee 100, 14469 Potsdam, Germany; cherrmann@atb-potsdam.de (C.H.); rschneider@atb-potsdam.de (R.S.); jvenus@atb-potsdam.de (J.V.)
5 Institute of Agricultural and Urban Ecological Projects, Berlin Humboldt University (IASP), Philippstr. 13, 10115 Berlin, Germany; patrice.ramm@iasp.hu-berlin.de
* Correspondence: mp@b3-bornim.de

**Abstract:** The aim of this study was to investigate the production of succinic acid from lactose concentrate, a by-product of cheese-making, using *Actinobacillus succinogenes* and *Basfia succiniciproducens*. Although the ability of these strains to metabolize different sugars is already known, their application in the conversion of lactose bears high potential for optimization. With regard to *B. succiniciproducens*, this approach is completely novel. In particular, the effect of the medium's sugar concentration as well as the ability of its supplementation with yeast extract to prevent a lack of essential nutrient proteins and vitamins was examined. Lactose-based media containing sugar concentrations of between 20 and 65 g L$^{-1}$ and 5 g L$^{-1}$ of yeast extract were fermented, with both strains showing comparable performances. The best results in terms of succinic acid yield and acid concentration—0.57 g g$^{-1}$ initial sugar and 23 g L$^{-1}$—were achieved at an initial sugar concentration of 43 g L$^{-1}$. The necessity of yeast extract was demonstrated using the sugar-optimized medium without supplementation. As a result, the yield and concentration of succinic acid dropped to 0.34 g g$^{-1}$ and 13 g L$^{-1}$ and the sugar consumption decreased from more than 99 to less than 55%. Therefore, the supplementation amount of 5 g L$^{-1}$ of yeast extract can be regarded as well-balanced.

**Keywords:** *Actinobacillus succinogenes*; *Basfia succiniciproducens*; succinic acid; lactose concentrate; yeast extract; platform chemical

## 1. Introduction

Succinic acid is a compound of high industrial interest, especially in fine chemistry, where it can be used as a platform chemical—for example, in the food and pharmaceutical industries [1]—or for the production of biopolymers [2]. The production of biopolymers based on succinic acid might be a good alternative to conventional plastics based on fossil oil [3].

In 2020, the annual global production of bio-based succinic acid was valued at 215 million USD, and this is expected to grow with a compound annual growth rate of 11% between 2021 and 2027 [4]. To satisfy the demand for bio-based succinic acid, large amounts of biomass are needed. With regard to the recent 'food versus fuel' discussion [5], organic residues and lignocellulosic biomass, representing the most abundant renewable organic sources on Earth [6], should be used for the production of bio-based chemicals such

as succinic acid. In recent publications, scientists have presented research on succinic acid production from the organic fraction of municipal solid waste [7], sugarcane bagasse [8], carob pods [9], corn stover [10], and straw [11] using the succinogenic bacterial strains *A. succinogenes* and *B. succiniciproducens*.

There are several metabolic pathways known to be capable of succinic acid synthesis that exist in different types of microorganisms. Succinic acid is a metabolite that typically occurs in all living organisms in the tricarboxylic acid cycle (TCA), where it is produced in the presence of oxygen (oxidative TCA). However, succinic acid can also be produced under anaerobic conditions utilizing the reductive TCA pathway [12]. The reductive pathway is a promising approach for succinic acid fermentation, as it allows the incorporation of carbon dioxide in degradation reactions with 6-carbon sugars [12]. Consequently, succinic acid fermentation could be used as a method to reduce industrial carbon dioxide emissions. The possibility of connecting industrially produced carbon dioxide and succinic acid fermentation has been demonstrated previously—for example, by [13,14].

In the past, succinic acid was synthesized by the hydrogenation of maleic anhydride, a chemical produced from fossil oil [12]. In contrast, the discovery and utilization of succinic acid-yielding bacteria, such as *A. succinogenes* [15] and *B. succiniciproducens* [16], allow the bio-based production of succinic acid by microbial fermentation, which is already applied on an industrial scale [12].

However, in spite of the existence of industrial-scale applications of succinogenic bacteria, several questions remain unsolved. In particular, the fermentation of substrates based on different biological residues bears high potential for optimization. A promising low-cost residue for microbial fermentation is whey, since its applicability has already been demonstrated [17–19]. Whey can also be effective in the production of protein concentrates, which are used as ingredients in sports beverages and special nutritional products [20]. The residue of the protein separation is a lactose-rich whey permeate. Concentrated by evaporation, this whey permeate turns into lactose concentrate. The use of lactose-based substrates, such as whey or lactose concentrate, offers the advantage that succinic acid fermentation with *A. succinogenes* is made possible without pretreatment by enzymatic hydrolysis because this strain is able to utilize glucose, galactose, and lactose [15]. Similar to *A. succinogenes*, *B. succiniciproducens* is capable of utilizing a diverse range of sugars— e.g., glucose, galactose, xylose, mannose, and sucrose [16]. Although the fermentation of multiple sugars by *B. succiniciproducens* is described in the literature, the authors of the present study are not aware of any studies dealing with it with respect to lactose.

With respect to *A. succinogenes* or *B. succiniciproducens*, an unbalanced ratio of essential nutrients could decrease succinic acid productivity. Yeast extract is often used for the supplementation of carbohydrate-rich media, since it is a source of various essential nutrients [8,19,21]. As yeast extract represents a significant expenditure, the reduction in its application to a minimum level is an important measure to reduce the production costs of succinic acid [22,23].

According to the modelling approach implemented by [24], an optimal synthetic medium for succinic acid fermentation should contain 84.6 g L$^{-1}$ glucose and 14.5 g L$^{-1}$ yeast extract. Consequently, the optimal initial glucose to yeast extract ratio is 5.8. This assumption is supported by findings published by [25], who observed a limited succinic acid production at an initial glucose to yeast extract ratio of 7.5; the initial sugar concentration in the study of [25] was 75 g L$^{-1}$. Although the present study was not focused on glucose but rather on lactose, the aforementioned information indicates that the supplementation with yeast extract might be able to improve a lactate-based medium too.

With respect to lactose-based medium, the dependencies between fermentation efficiency and nutrient composition as well as the optimal ratio of mono- and disaccharides present in the substrate have not been discussed in detail so far; the presented work aims to close this gap.

The main objective of the present study was to investigate the suitability of lactose concentrate as a feedstock for the two natural acid producers *A. succinogenes 130Z* (DSM

22257) and *B. succiniciproducens* (DSM 22022). In this was, fermentation tests were performed in batch mode using lactose concentrate generated from cheese whey as a feedstock and main compound for the fermentation media. The present study consisted of two experiments. During the first experiment, the influence of the sugar concentration on the succinic acid productivity was assessed by applying sugar concentrations between 20 and 65 g L$^{-1}$. Concerning the low protein content of the lactose concentrate and presumable lack of vitamins, yeast extract was used for the supplementation. In the second experiment, the necessity of the supplementation with yeast extract to increase the succinic acid yield was investigated. Based on the two described experimental approaches, the present study had the following objectives:

- Determining the potential of a by-product from cheese-making, lactose concentrate, as a feedstock for bio-based succinic acid production.
- Comparing the succinic acid fermentation performance of *A. succinogenes 130Z* (DSM 22257) and *B. succiniciproducens* (DSM 22022).
- Examining the effect of yeast extract in a lactose-based medium on succinic acid production.

## 2. Materials and Methods

### 2.1. Microbial Strains and Precultivation

*A. succinogenes 130Z* (DSM 22257) and *B. succiniciproducens* (DSM 22022) were procured from the DSMZ-German Collection of Microorganisms and Cell Cultures GmbH (Braunschweig, Germany). The bacterial strains were kept as cryogenic stocks at −80 °C in 50% glycerol aqueous solution. Precultures were produced by the inoculation of 100 mL tryptic soy broth (Merck KGaA, Darmstadt, Germany) in shaker flasks within 24 h at 37 °C in aerobic conditions. The used orbital shaker had an agitation speed of 150 rpm. The flasks were heat-sterilized at 121 °C for 15 min.

### 2.2. Fermentation Media

Lactose concentrate, a concentrate from whey permeate and a by-product from cheese-making (Karwendel-Werke-Huber GmbH & Co. KG, Buchloe, Germany), and yeast extract (Ohly KAT, Deutsche Hefewerke GmbH, Nürnberg, Germany) were used for the preparation of different fermentation media. Lactose concentrate is currently used as a substrate in biogas plants. To save transportation costs, it was concentrated by the Karwendel-Werke-Huber GmbH to a dry matter content of about 30%. For the preparation of the media, the lactose concentrate was diluted with water to adjust three different target sugar concentrations of 20, 40, and 60 ± 5 g L$^{-1}$; the resulting media were called A, B, and C, respectively. The media used for the first experiment were enriched with yeast extract at a concentration of 5 g L$^{-1}$. The dosage of yeast extract was indicated by the letter Y; consequently, these media were called $A_Y$, $B_Y$, and $C_Y$. Lactose concentrate and yeast extract were autoclaved separately at 121 °C for 15 min and then aseptically mixed. The second experiment was conducted with medium B, whereby the addition of yeast extract was omitted. Each fermentation was started by adding 100 mL of preculture to 900 mL of medium.

### 2.3. Experimental Setup and Operation

Fermentation was carried out in batch mode utilizing a 3 L Biostat bioreactor system (Sartorius AG, Göttingen, Germany). As optimal growth conditions, the literature states a mesophilic temperature (37 °C) [15,16] and pH values of 6.8 [9,21] or 6.5 [14] for both strains. Based on this information, a temperature of 37 °C and a pH value of 6.7 were applied. The pH value was measured with the sensor EasyFerm Plus PHI K8 225 (Hamilton Bonaduz AG, Bonaduz, Switzerland) and automatically regulated by the addition of 5 N of NaOH. Carbon dioxide was continuously sparged into the reactor with a flow rate of 0.2 L min$^{-1}$. A double Rushton turbine was used for stirring with an agitation speed of 300 rpm. The initial volume of each fermentation batch, consisting of medium and preculture, was 1 L. The impact of the initial sugar concentration on the production of

succinic acid and by-products was examined during the first experiment. Therefore, a series of fermentation tests with either *A. succinogenes* or *B. succiniciproducens* was conducted with fermentation media $A_Y$, $B_Y$, and $C_Y$.

In a second experimental approach, we investigated if a decrease in nitrogen in the medium affects the bacterial sugar consumption and hence the acid production. Therefore, fermentations were repeated with medium B without the addition of yeast extract. The test duration of the second experiment amounted to 55 h.

During all fermentations, the sampling of the fermentation medium was carried out regularly in order to characterize the production kinetics of succinic acid and by-products, such as acetic, formic, and lactic acid. Moreover, the amount and composition of residual saccharides were also analyzed. Upon each sampling, the succinic acid-producing bacteria were inactivated by heating the sample for 20 min at 95 °C.

### 2.4. Analytical Methods

The concentrations of sugars and acids were obtained by high-performance liquid chromatography (HPLC) using a ULTIMATE 3000 system (Thermo Fischer Scientific, Waltham, MA, USA) with an Eurokat H column (300 mm × 8 mm × 10 μm, Knauer, Berlin, Germany) and a refractive index detector (Shodex RI-101, Showa Denko, Tokio, Japan). The temperature of the column thermostat was 40 °C; the eluent, 0.01 N sulfuric acid, was pumped through the column at a speed of 0.8 mL min$^{-1}$. The peak areas and retention times were compared to those of standard solutions. Before the samples of lactose concentrate and fermentation medium were analyzed by HPLC, they were diluted with water. The dilution factor of the samples was determined by the present concentration of the analyzed sugars (glucose, galactose, saccharose, and lactose) and acids (succinic, acetic, formic, and lactic acid). The achieved concentration of every sugar and acid component had to fit the measuring range of between 0.5 and 5.0 g L$^{-1}$. The HPLC analysis of the yeast extract required pretreatment by protein precipitation with Carrez reagent, as described by [26]. The determination of the total Kjeldahl nitrogen content (TKN) was carried out employing a standard method [27]. The pH value was measured according to the methods stated by [28,29]. The protein content was calculated from the TKN by multiplication by a factor of 6.25 [28].

### 2.5. Calculation of the Fermentation Parameters

The results retrieved from the fermentation experiments were evaluated by comparing them with the determined final succinic acid concentration and the calculated succinic acid yield with regard to the initial sugar concentration. The formation of by-products served as an additional important parameter. The synthesis of the by-products was expressed as mass ratios between succinic acid and acetic, formic, or lactic acid. The total concentration of sugar was determined as the sum of mono- and disaccharides and given as the equivalent of glucose. The sugar consumption was defined as the amount of sugar converted in relation to the initial amount of sugar. The loss of medium due to sampling was taken into account for the calculation of the acid yields, the mass ratio between succinic acid and by-products, and the sugar consumption.

## 3. Results and Discussion

### 3.1. Feedstock and Fermentation Media

The chemical characteristics of the applied lactose concentrate as well as the supplemented yeast extract, analyzed in duplicates, are given in Table 1.

The lactose concentrate had a sugar concentration of 268 g kg$^{-1}$ with a lactose percentage of almost 90% and a protein content of 12 g kg$^{-1}$. In comparison to the lactose concentrate, the protein content of the yeast extract was sixty times higher and had a value of 723 g kg$^{-1}$. The average initial sugar concentrations of the media enriched with yeast extract ($A_Y$, $B_Y$, and $C_Y$) were 21, 43, and 64 g L$^{-1}$ and the maximum standard deviation was 1 g L$^{-1}$. The initial protein concentration of the media ranged between 4.5 and 6.5 g L$^{-1}$.

About 56 to 79% of the protein content was supplied by the yeast extract, depending on its ratio. The mass ratio of the initial sugar to yeast extract of the media amounted to 4, 8 or 9 and 13 ($A_Y$, $B_Y$, and $C_Y$, cf. Table 2). A limited succinic acid production can be expected for media showing an initial sugar to yeast extract ratio higher than 7.5, as described in the introduction. Consequently, limited succinic acid production and sugar consumption were predicted for media $B_Y$ and $C_Y$.

**Table 1.** Chemical characteristics of lactose concentrate and yeast extract.

| Parameter | Lactose Concentrate | Yeast Extract |
|---|---|---|
| Sugar (g kg$^{-1}$) [a] | $268.4 \pm 0.1$ | $6.6 \pm 0.2$ |
| Glucose (g kg$^{-1}$) | $7.2 \pm 0.1$ | n.d. |
| Galactose (g kg$^{-1}$) | $21.2 \pm 0.0$ | n.d. |
| Saccharose (g kg$^{-1}$) [a] | n.d. | $6.6 \pm 0.2$ |
| Lactose (g kg$^{-1}$) [a] | $240.0 \pm 0.0$ | n.d. |
| Protein (g kg$^{-1}$) | $12.0 \pm 0.0$ | $722.5 \pm 13.8$ |
| Lactic acid (g kg$^{-1}$) | $37.6 \pm 0.0$ | $24.3 \pm 0.7$ |

n. d., not detected; [a], regarded as glucose equivalents.

**Table 2.** Results of the succinic acid production by *A. succinogenes* and *B. succiniciproducens* from lactose concentrate.

| Parameter\Medium | *A. succinogenes* | | | | *B. succiniciproducens* | | | |
|---|---|---|---|---|---|---|---|---|
| | $A_Y$ | $B_Y$ | $C_Y$ | B | $A_Y$ | $B_Y$ | $C_Y$ | B |
| Initial sugar concentration (g L$^{-1}$) | 21 | 43 | 63 | 38 | 20 | 42 | 65 | 40 |
| Initial sugar to yeast extract ratio | 4 | 9 | 13 | - | 4 | 8 | 13 | - |
| Fermentation duration (h) | 19 | 46 | 47 | 55 | 20 | 46 | 47 | 54 |
| Final succinic acid concentration (g L$^{-1}$) | 13.0 | 22.8 | 24.0 | 13.3 | 12.4 | 21.6 | 19.5 | 13.3 |
| Yield of succinic acid (g g$^{-1}$) | 0.65 | 0.57 | 0.41 | 0.34 | 0.64 | 0.54 | 0.33 | 0.34 |
| Residual sugar concentration (g L$^{-1}$) | 0.2 | 0.3 | 12.9 | 16.1 | 0.0 | 0.2 | 17.8 | 17.9 |
| Sugar consumption (%) | 99.2 | 99.8 | 75.6 | 53.8 | 100.0 | 99.7 | 68.1 | 50.9 |
| Succinic acid to acetic acid ratio | 1: 0.43 | 1: 0.40 | 1: 0.42 | 1: 0.37 | 1: 0.35 | 1: 0.30 | 1: 0.32 | 1: 0.29 |

The average initial concentration of sugar in the yeast extract-free medium B amounted to 39 g L$^{-1}$. As for media $B_Y$ and $C_Y$, a decline in succinic acid yield was also very likely for medium B. The lactose concentrate and yeast extract contained lactic acid in concentrations of 37.6 g kg$^{-1}$ and 24.3 g kg$^{-1}$, as shown in Table 1. Consequently, the average initial contents of lactic acid in the media $A_Y$, $B_Y$, B, and $C_Y$ were 3, 6, 5, and 9 g L$^{-1}$.

### 3.2. Optimizing the Sugar Concentration for A. succinogenes and B. succiniciproducens

The results of the fermentation experiments with media $A_Y$, $B_Y$, and $C_Y$ are summarized in Table 2 for both *A. succinogenes* and *B. succiniciproducens*. The highest yields of succinic acid were achieved with an average initial sugar concentration of 21 g L$^{-1}$ (medium $A_Y$). In that case, the succinic acid production was comparable for both strains, showing yields of 0.65 g (*A. succinogenes*) and 0.64 g (*B. succiniciproducens*) per g of initial sugar. The application of media $B_Y$ or $C_Y$ resulted in small differences in the succinic acid yields of both strains. *A. succinogenes* achieved values of 0.57 (medium $B_Y$) and 0.41 (medium $C_Y$) g succinic acid per g initial sugar, while the *B. succiniciproducens* yields amounted to 0.54 (medium $B_Y$) and 0.33 (medium $C_Y$) g g$^{-1}$. In comparison to media $A_Y$, the yields were 14% and 43% lower for media $B_Y$ and $C_Y$.

The final succinic acid concentrations of 13.0 g L$^{-1}$ (*A. succinogenes*) and 12.4 g L$^{-1}$ (*B. succiniciproducens*) were measured at the end of the fermentation trials with medium $A_Y$. The increase in the average initial sugar concentration from 21 g L$^{-1}$ (medium $A_Y$) to 43 g$^{-1}$ (medium $B_Y$) strongly enhanced the succinic acid production with respect to its final concentration, but at the expense of the yield. After 46 h of fermentation with medium $B_Y$, the concentrations of succinic acid reached 22.8 g L$^{-1}$ for *A. succinogenes* and

21.6 g L$^{-1}$ for *B. succiniciproducens*, respectively. The course of succinic acid production during fermentation is exemplarily shown for *A. succinogenes* in Figure 1.

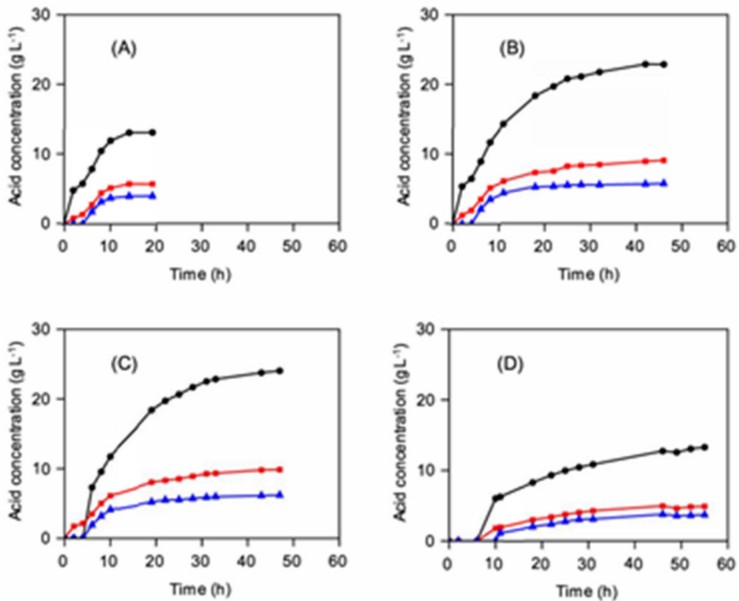

**Figure 1.** Acid production during fermentation with *A. succinogenes*: medium A$_Y$ (**A**), medium B$_Y$ (**B**), medium C$_Y$ (**C**), medium B (**D**); concentration of succinic acid (black, circle), acetic acid (red, square), and formic acid (blue, triangle).

The further increase in the average initial sugar concentration to 64 g L$^{-1}$ (medium C$_Y$) was not effective, since it caused only a small increase (*A. succinogenes*) or even a small decrease (*B. succiniciproducens*) in the succinic acid production compared to medium B$_Y$. This observation was accompanied by a sharp decrease in the succinic acid yield from 0.65 to 0.41 g g$^{-1}$ initial sugar (*A. succinogenes*) and from 0.64 to 0.33 g g$^{-1}$ initial sugar (*B. succiniciproducens*) compared to medium A$_Y$. Although not tested with lactose, a similar effect was observed for glucose by [25]. In the respective study, a succinic acid concentration of 41 g L$^{-1}$ was achieved by the fermentation of a synthetic medium with a glucose concentration of 51 g L$^{-1}$; the further increase in the initial glucose concentration to 74 g L$^{-1}$ did not enhance the succinic acid production [25]. Besides the negative effects on acid productivity caused by the high initial sugar concentration, a decrease in succinic acid yield might also be explained by the inhibitory effect of the acids produced during fermentation. The author of [30] reported that the succinic acid production stopped when the concentration of the microbially produced acid mixture reached a value of 45 g L$^{-1}$. With values of 49 g L$^{-1}$ (*A. succinogenes*) and 44 g L$^{-1}$ (*B. succiniciproducens*) found in the present study, the fermentation of medium C$_Y$ resulted in concentrations in a similar range as that described by [31]. Therefore, the lowered yield might have occurred due to product inhibition.

With regard to the succinic acid production potential, the yields observed in this study during the fermentation of lactose concentrate are in a similar range as the yields from other organic by-products that are listed in the literature. The authors of [8] obtained a final succinic acid concentration of 23 g L$^{-1}$ from hydrolyzed sugarcane bagasse (supplemented with yeast extract and minerals), where the initial sugar concentration (xylose) was 52 g L$^{-1}$. Meanwhile, the authors of [19] measured 28 g L$^{-1}$ of succinic acid after 48 h of fermentation using a medium with an initial cheese whey concentration of 100 g L$^{-1}$. The authors of [7] published a techno-economic case study of succinic acid production from the organic fraction of municipal waste and calculated profitability indicators. The calculation was based on a succinic acid productivity of 0.89 g L$^{-1}$ h$^{-1}$ and a production capacity higher than 40,000 t per year. From medium B$_Y$, a similar succinic acid productiv-

ity of 0.87 g L$^{-1}$ h$^{-1}$ was achieved during 23 h of fermentation with *A. succinogenes*; the corresponding succinic acid yield was 0.50 g g$^{-1}$. *B. succiniciproducens* reached the same productivity after 20 h of fermentation, but there was a small decline in the succinic acid yield to 0.45 g g$^{-1}$.

During the fermentation of media A$_Y$ and B$_Y$, more than 99% of the sugars were converted. When applying medium B$_Y$, no decline in the sugar consumption occurred. This is surprising, as in comparison to the results obtained with medium A$_Y$ the conditions were less favorable with respect to the initial sugar to yeast extract ratio (increase from 4 to 8 or 9).

In contrast to the efficient sugar conversion achieved with medium B$_Y$, the sugar consumption decreased to less than 76% during the fermentation of medium C$_Y$. At the end of the fermentation trials, the (residual) sugar concentrations in medium C$_Y$ were 13 g L$^{-1}$ (*A. succinogenes*) and 18 g L$^{-1}$ (*B. succiniciproducens*), with galactose percentages of 81% and 80%. Therefore, it may be assumed that both bacteria strains were capable of cleaving lactose into the components glucose and galactose. Subsequently, glucose was rapidly utilized and the consumption of galactose was likely limited. This assumption is supported by results from previous studies. For example, while conducting succinic acid fermentations with media showing initial whey concentrations of 50 to 100 g L$^{-1}$ and supplemented with minerals, peptone, and yeast extract, the authors of [19] observed a fast consumption of glucose but a delay in the consumption of galactose, especially during the first 24 h.

Apart from succinic acid, the main by-product of the succinic acid fermentations was acetic acid. The production of 1 g succinic acid with *A. succinogenes* from media A$_Y$, B$_Y$, and C$_Y$ was accompanied by the production of 0.42 g of acetic acid on average (Table 2). In comparison to that, formic and lactic acid were formed at ratios of 0.27 g and 0.03 g per g succinic acid.

The formation of lactic acid occurred, especially during the fermentation of medium C$_Y$ with *B. succiniciproducens*. The final lactic acid concentration was 12 g L$^{-1}$ and the final mass ratio between succinic and lactic acid reached a value of 1: 0.26. In comparison, after the fermentation of medium C$_Y$ with *A. succinogenes*, the ratio was 1: 0.06. The fermentation of a synthetic medium containing 60 g L$^{-1}$ of glucose and 6 g L$^{-1}$ of yeast extract with *B. succiniciproducens* conducted by [10] resulted in a comparable lactic acid concentration of 10 g L$^{-1}$, while the final mass ratio between succinic and lactic acid was 1: 0.35.

### 3.3. Yeast Supplementation for Optimization of Succinic Acid Fermentation

The aim of the second experiment was to find out if the supplementation with yeast extract had a positive influence on the succinic acid production. To demonstrate the importance of the essential nutrients contained in yeast extract, the experiment evaluated in Section 3.2 was repeated without adding yeast extract (medium B). A sugar concentration of about 40 g per liter was applied, as this was determined as optimal from the results of the previous experiment.

The fermentations with *A. succinogenes* and *B. succiniciproducens* led to identical succinic acid yields of 0.34 g g$^{-1}$ of initial sugar (Table 2). Within 55 h, the succinic acid concentration reached a value of 13.3 g L$^{-1}$. Thus, the succinic acid production with medium B was about 40% lower compared to the same medium supplemented with yeast extract (B$_Y$). Moreover, the sugar consumption decreased sharply from more than 99% to 54% (*A. succinogenes*) and 51% (*B. succiniciproducens*), respectively.

Similar to the fermentation of medium C$_Y$, the degradation of galactose was limited during the fermentation of medium B. The residual sugar concentrations were 16 and 18 g L$^{-1}$ with galactose shares of 75% (*A. succinogenes*) and 67% (*B. succiniciproducens*). The negative effect of the lack of nutrients is exemplarily shown for *A. succinogenes* in Figure 2.

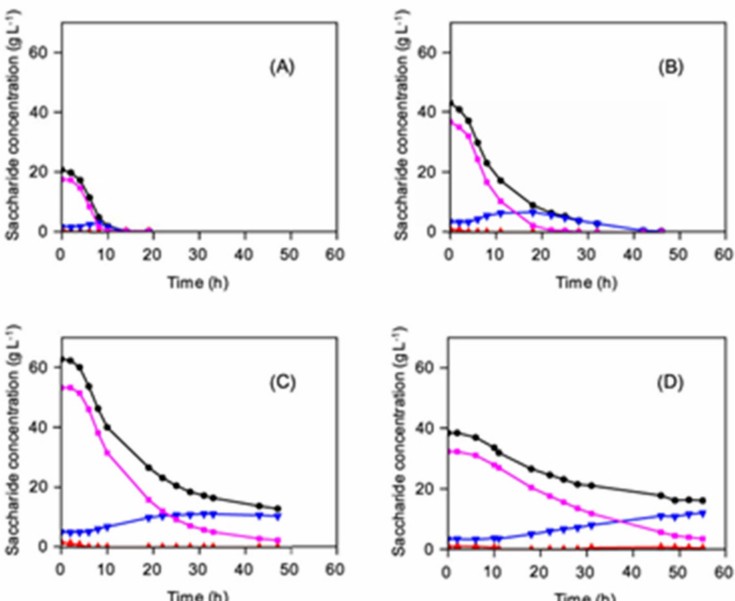

**Figure 2.** Sugar degradation during fermentation with *A. succinogenes*: medium $A_Y$ (**A**), medium $B_Y$ (**B**), medium $C_Y$ (**C**), medium B (**D**); concentration of sugar (black, circle,), lactose (violet, square), glucose (red, upward-pointing triangle), and galactose (blue, downward-pointing triangle).

The results indicate a lack of essential nutrients contained in yeast extract. The authors of [31] investigated the supplementation of the medium with yeast extract during the fermentation of high-sugar-content beverages with *A. succinogenes*. The concentration of yeast extract was $5 \text{ g L}^{-1}$, the initial sugar concentration varied between 46 and $56 \text{ g L}^{-1}$. The omission of yeast extract resulted in a decrease in the fermentation performance depending on the medium (pineapple juice or ACE juice) by 14% or 13% (final concentration of acid mixture), 21% or 16% (sugar consumption), and 3% or 14% (succinic acid concentration), respectively. Therefore, the importance of yeast extract for the fermentation of sugars is clearly recognizable in [31]; this supports the results of the present study.

### 3.4. Different Microbial Strains for Succinic Acid Fermentation

Comparing the results of the present study obtained with *A. succinogenes* and *B. succiniciproducens* to those of other research articles on succinic acid fermentation, it becomes clear that there are multiple other strains that can be used for this purpose. For example, it was shown that *Mannheimia succiniciproducens*, a strain that is genetically close to *B. succiniciproducens*, is also able to ferment lactose efficiently [32]. The fact that these two strains showed a similar fermentation performance increases the trustworthiness of our results and shows reproducibility [33]. Another strain that has been described in the literature as particularly suitable is *Enterobacter aerogenes LU2* [34]. The authors of the mentioned publication showed that *E. aerogenes* is also able to utilize lactose; in their experiments, succinic acid concentrations of up to $57.7 \text{ g L}^{-1}$ were achieved. With succinic acid yields of between 12 and $24 \text{ g L}^{-1}$, much lower concentrations were achieved with *A. succinogenes* and *B. succiniciproducens* in the present study. However, it must be considered that the sugar content of the growth medium differed clearly. With up to $140 \text{ g L}^{-1}$, the sugar concentration applied by [34] was more than twice as high as that used in the experiments presented here. Moreover, the fermentation with *E. aerogenes* was much slower, taking approximately 144 h to terminate. The presented fermentation experiments with *A. succinogenes* and *B. succiniciproducens* finished in less than 50 h. To reach a similar velocity with *E. aerogenes*, the authors of the respective study had to lower the lactose concentration to $60 \text{ g L}^{-1}$, which is in a similar range as the results presented here [34].

## 4. Conclusions

Lactose concentrate from cheese production is a very suitable feedstock for succinic acid production using *A. succinogenes* or *B. succiniciproducens.* Comparing both strains to *Mannheimia succiniciproducens* and *Enterobacter aerogenes LU2*, we found that they appeared to be very efficient lactose users as well. A broad range of initial sugar concentrations from 20 to 65 g L$^{-1}$ in the medium can be applied. The best performance was achieved at an initial sugar concentration of 43 g L$^{-1}$ for both the acid yield per initial sugar, at 0.57 g per g initial sugar, and the acid concentration, at up to 23 g L$^{-1}$. The supplementation with yeast extract appeared to be mandatory, since its absence resulted in lower succinic acid concentrations. A yeast extract concentration of 5 g L$^{-1}$ proved to be more than sufficient. Further investigations into the minimal concentration of yeast extract in the medium are recommended to increase the profitability of the process.

**Author Contributions:** Conceptualization, C.T. and J.L.; methodology, C.T., J.L., and R.S.; validation, C.T., M.P., and J.V.; formal analysis, C.T. and P.R.; investigation, C.T., J.L., and R.S.; resources, M.P., C.H., and J.V.; data curation, C.T. and R.S.; writing—original draft preparation, C.T., C.A., and P.R.; writing—review and editing, C.T., C.A., C.H., P.R., J.V., and M.P.; visualization, C.T. and P.R.; supervision, M.P.; project administration, M.P.; funding acquisition, C.A. and M.P. All authors have read and agreed to the published version of the manuscript.

**Funding:** The research was funded by the Bundesministerium für Wirtschaft und Energie (BMWi) within the framework of the program "Zentrales Innovationsprogramm Mittelstand (ZIM)", grant numbers 16KN070127, 16KN070126 and 16KN070128.

**Informed Consent Statement:** Not applicable.

**Data Availability Statement:** The reported results are available upon request to the authors.

**Acknowledgments:** We give our special acknowledgements to the members of the chemical-analytical lab of the Leibniz Institute for Agricultural Engineering and Bioeconomy for their ambitious support. Finally, we want to thank Olaf Luschnig from the BioEnergie Verbund e.V. for his organizational support.

**Conflicts of Interest:** The authors declare no conflict of interest.

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
