# Peer review of "Influence of the Initial Sugar Concentration and Supplementation with Yeast Extract on Succinic Acid Fermentation in a Lactose-Based Medium"

_fermentation, doi:10.3390/fermentation7040221_

Round 1

Reviewer 1 Report

The manuscript entitled ‘Influence of the initial sugar concentration and supplementation with yeast extract on the succinic acid fermentation from lactose based medium’ satisfies the scope of the journal of Fermentation. The aims of the study are clearly demonstrated, but the approach is very simple. You should implement into the research the optimization of the concentration of yeast extract and would be worthy to define the interaction of the concentration of saccharides and yeast extract. The introduction provides a very good overview of the current knowledge on succinic acid fermentation. The references contain the important papers dealing with research efforts of the strain improvement and optimization of fermentation environment, moreover the industrial production and the application of succinic acid. The description of the methods applied is exact and satisfactory. The gained results are clearly presented, and the conclusion is supported by them.

Author Response

The manuscript entitled ‘Influence of the initial sugar concentration and supplementation with yeast extract on the succinic acid fermentation from lactose based medium’ satisfies the scope of the journal of Fermentation. The aims of the study are clearly demonstrated, but the approach is very simple.

We are pleased that the content and goals of the work were clear and comprehensible to you.

You should implement into the research the optimization of the concentration of yeast extract and would be worthy to define the interaction of the concentration of saccharides and yeast extract.

In the past, other authors investigated optimal ratios between glucose and yeast. To show that yeast improves the digestibility of lactose concentrate as well, we used a similar amount of yeast as in previous studies with glucose. We have now rephrased the section from page 2, line 93 to make this clearer:

 According to a modelling approach performed by [25], an optimal synthetic medium for succinic acid fermentation should contain 84.6 g L-1 glucose and 14.5 g L-1 yeast extract. Consequently, the optimal initial glucose to yeast extract ratio is 5.8. This assumption is supported by findings published by [26], who observed a limited succinic acid production at an initial glucose to yeast extract ratio of 7.5; the initial sugar concentration in the study from [26] was 75 g L-1. Although the present study was not focused on glucose, but on lactose, the aforementioned information indicates that the supplementation with yeast extract might improve a lactate based medium too.

The introduction provides a very good overview of the current knowledge on succinic acid fermentation. The references contain the important papers dealing with research efforts of the strain improvement and optimization of fermentation environment, moreover the industrial production and the application of succinic acid. The description of the methods applied is exact and satisfactory. The gained results are clearly presented, and the conclusion is supported by them.

We are grateful that you acknowledge the quality of the present work.

Reviewer 2 Report

Manuscript called "Influence of the initial sugar concentration and supplementation with yeast extract on the succinic acid fermentation from lactose based medium" contains results of research concerning succinic acid production exploiting 2 well known succinic acid producers Actinobacillus succinogenes (eg. Guettler MV, Rumler D, Jain MK. Actinobacillus succinogenes nov., a novel succinic-acid-producing strain from the bovine rumen. Int. J. Syst. Bacteriol. 1999; 49:207–16) and Basfia succiniciproducens (eg. Scholten E, Dägele D. Succinic acid production by a newly isolated bacterium. Biotechnol. Lett. 2008; 30:2143–2146). Results described in the manuscript comes from scientific experiments concerning microbial cultivations on fermentation media containing lactose concentrate which is a by-product from cheese making. The authors focused on optimization of sugar concentration and yeast extract supplementation in fermentation media. Despite appropriate research design and adequately described methods, presented results lack novelty and are of poor significance for potential readers interested in succinic acid biotechnology. You can find previously published scientific articles concerning succinic acid production from cheese whey using Actinobacillus succinogenes (Wan C, Li Y, Shahbazi A, Xiu S. Biochem. Biotechnol. 2008; 145:111-119) and necessity of fermentation media supplementation with organic extracts because of Actinobacillus  auxotrophic character (McKinlay JB, Laivenieks M, Schindler BD, McKinlay AA, Siddaramappa S, Challacombe JF, Lowry SR, Clum A, Lapidus AL, Burkhart KB, Harkins V, Vieille C. A genomic perspective on the potential of Actinobacillus succinogenes for industrial succinate production. BMC Genomics 2010; 11:680). Although the lack of scientific reports about Basfia succiniciproducens cultivations on lactose containing media you can find some describing Mannheimia succiniciproducens  (eg. Lee PC, Lee SY, Hong SH, Chang HN. Batch and continuous cultures of Mannheimia succiniciproducens MBEL55E for the production of succinic acid from whey and corn steep liquor. Bioprocess Biosyst Eng. 2003;26(1):63–7.) which is almost identical to Basfia succiniciproducens in genome sequence (Ahn, J.H., Seo, H., Park, W. et al. Enhanced succinic acid production by Mannheimia employing optimal malate dehydrogenase. Nat Commun 11, 1970 (2020). https://doi.org/10.1038/s41467-020-15839-z). Moreover values of succinic acid yield and concentration in postfermentation media described  in the manuscript are not higher than previously published results for Actinobacillus and Basfia not to mention Enterobacter strains (Szczerba H, KomoÅ„-Janczara E, Dudziak K, WaÅ›ko A, TargoÅ„ski Z. A novel biocatalyst, Enterobacter aerogenes LU2, for efficient production of succinic acid using whey permeate as a cost-effective carbon source. Biotechnol Biofuels. 2020;13:96. Published 2020 May 29. doi:10.1186/s13068-020-01739-3).

In conclusion I recommend to reject reviewed manuscript. Its hypothetical publication in Fermentation Journal should be possible only after thorough improvement of results novelty or succinic acid production parameters.

Author Response

Manuscript called "Influence of the initial sugar concentration and supplementation with yeast extract on the succinic acid fermentation from lactose based medium" contains results of research concerning succinic acid production exploiting 2 well known succinic acid producers Actinobacillus succinogenes (eg. Guettler MV, Rumler D, Jain MK. Actinobacillus succinogenes nov., a novel succinic-acid-producing strain from the bovine rumen. Int. J. Syst. Bacteriol. 1999; 49:207–16) and Basfia succiniciproducens (eg. Scholten E, Dägele D. Succinic acid production by a newly isolated bacterium. Biotechnol. Lett. 2008; 30:2143–2146). Results described in the manuscript comes from scientific experiments concerning microbial cultivations on fermentation media containing lactose concentrate which is a by-product from cheese making. The authors focused on optimization of sugar concentration and yeast extract supplementation in fermentation media.

You recorded the content of the work well and summarized it appropriately. To the summary of the expert, we would like to add that Actinobacillus succinogenes and Basfia succiniciproducens have indeed long been known and well characterized, but that their substrate-specific behavior has not yet been adequately researched, which is necessary for industrial use. This is one of the innovative highlights of the present work.

Your criticism is mainly based on the degree of novelty of the work. We therefore assume that you agree with the structure and methodology of the present work. We have now emphasized the degree of innovation in the work more clearly (see comments below).

Despite appropriate research design and adequately described methods, presented results lack novelty and are of poor significance for potential readers interested in succinic acid biotechnology. You can find previously published scientific articles concerning succinic acid production from cheese whey using Actinobacillus succinogenes (Wan C, Li Y, Shahbazi A, Xiu S. Biochem. Biotechnol. 2008; 145:111-119) and necessity of fermentation media supplementation with organic extracts because of Actinobacillus auxotrophic character (McKinlay JB, Laivenieks M, Schindler BD, McKinlay AA, Siddaramappa S, Challacombe JF, Lowry SR, Clum A, Lapidus AL, Burkhart KB, Harkins V, Vieille C. A genomic perspective on the potential of Actinobacillus succinogenes for industrial succinate production. BMC Genomics 2010; 11:680).

We agree with you that there is already extensive literature on the named species. you also mention articles through which conclusions are drawn about the metabolism of lactose. In our experience, however, this is not sufficient to justify approaches on an industrial scale. The present results allow yield estimates for lactose-rich substrates, which have so far been neglected on an industrial scale, with greater accuracy. We therefore assume that the present results will contribute to greater substrate diversity for the future industrial production of succinic acid.

Although the lack of scientific reports about Basfia succiniciproducens cultivations on lactose containing media you can find some describing Mannheimia succiniciproducens (eg. Lee PC, Lee SY, Hong SH, Chang HN. Batch and continuous cultures of Mannheimia succiniciproducens MBEL55E for the production of succinic acid from whey and corn steep liquor. Bioprocess Biosyst Eng. 2003;26(1):63–7.) which is almost identical to Basfia succiniciproducens in genome sequence (Ahn, J.H., Seo, H., Park, W. et al. Enhanced succinic acid production by Mannheimia employing optimal malate dehydrogenase. Nat Commun 11, 1970 (2020). https://doi.org/10.1038/s41467-020-15839-z).

We thank you for pointing out the article on Mannheimia succiniciproducens, which we have now included as an additional source. Since the strains examined are not the same strains as M. succiniproducens despite their genetic similarity, we see a novelty value here. The fact that there are already results for a similar strain actually increases the trustworthiness of both articles and increases reproducibility. 

We have now discussed the reference suggested by the reviewer in the manuscript as follows:

Page 9, Line 353: “Comparing the results of the present study obtained with A. succinogenes and B. succiniciproducens to other research articles on succinic acid fermentation, it becomes clear that there are multiple other strains, which can be used for this purpose. For example, it was shown that Mannheimia succiniciproducens, a strain that is genetically close to Basfia succiniciproducens, is also able the ferment lactose efficiently [34]. The fact that these two strains showed a similar fermentation performance increases the trustworthiness and shows reproducibility [35].

Moreover values of succinic acid yield and concentration in postfermentation media described in the manuscript are not higher than previously published results for Actinobacillus and Basfia not to mention Enterobacter strains (Szczerba H, Komoń-Janczara E, Dudziak K, Waśko A, Targoński Z. A novel biocatalyst, Enterobacter aerogenesLU2, for efficient production of succinic acid using whey permeate as a cost-effective carbon source. Biotechnol Biofuels. 2020;13:96. Published 2020 May 29. doi:10.1186/s13068-020-01739-3).

Again, we would like to thank you for the attached source, which we are now discussing in the present manuscript. The fact that both, the article on Mannheimia succiniciproducens and the article on Enterobacter aerogenes, were published in 2020 once again illustrates the great interest of the research community in this topic. With the here presented knowledge on Actinobacillus succinogenes and Basfia succiniciproducens, the range of suitable strains to produce succinic acid from lactose-rich substrates is considerably expanded.

We have now discussed the reference suggested by the reviewer in the manuscript as follows:

Page 9, Line 353: “Comparing the results of the present study obtained with A. succinogenes and B. succiniciproducens to other research articles on succinic acid fermentation, it becomes clear that there are multiple other strains, which can be used for this purpose. For example, it was shown that Mannheimia succiniciproducens, a strain that is genetically close to Basfia succiniciproducens, is also able to ferment lactose efficiently [34]. The fact that these two strains showed a similar fermentation performance increases the trustworthiness and shows reproducibility [35]. Another strain, which has been described in literature as particularly suitable is Enterobacter aerogenes LU2 [36]. The authors of the mentioned publication have shown that E. aerogenes is also able to utilize lactose, in their experiments succinic acid concentrations up to 57.7 g L-1 were achieved. With succinic acid yields between 12 and 24 g L-1, much lower concentrations were achieved with A. succinogenes and B. succiniciproducens in the present study. However, it must be considered that the sugar content of the growth medium differed clearly. With up to 140 g L-1, the sugar concentration applied by [36] was more than twice as high as in the experiments presented here. Moreover, the fermentation with E. aerogenes was much slower, needing approximately 144 h to terminate. The here presented fermentation experiments with A. succinogenes and B. succiniciproducens finished in less than 50 h. To reach a similar velocity with E. aerogenes, the authors of the respective study had to lower the lactose concentration to  60 g L-1, which is in a similar range as the here presented results [36].

In conclusion I recommend to reject reviewed manuscript. Its hypothetical publication in Fermentation Journal should be possible only after thorough improvement of results novelty or succinic acid production parameters.

We understand your assessment. In accordance with the recommendation of the expert, we have emphasized the degree of novelty of the article more clearly. The sources suggested by you, which we have all incorporated into the present manuscript, were particularly helpful at this point (as discussed above). We would like to take this opportunity to thank you again for making a significant contribution to improving the manuscript.

To better highlight the innovation of the presented work, it has been related in more detail to recent publications that are addressing succinic acid producers (see answer to the previous comment).

Reviewer 3 Report

Abstract

Lines 18-19: ".... lactose concentrate, ....". Do you think about whey permeate?

Introduction

Lines 36: Succinic acid is very rarely mentioned as VFA.

Lines 41-43: Please provide more up-to-date data. 

Lines 47-50: Did you perform experiments using municipal solid wastes, sugarcane bagasse, carob pods and so on in this paper? Because that's what it sounds like.

Lines 89-94: I think that this fragment is not necessary cause the present study is regarding to lactose.

Lines 112-116: I think that whey permeate is a rich source of amino acids and trace metals. Hence, I believe that the use of yeast extract does not have to be explained by the fact that the whey permeate is poor. I believe also that the use of YE is fully justified in the preliminary research and the authors do not have to justify that its limitation is needed.

Lines 95-100: This information is no needed, as there is no experiments on elimination of side products in this paper. I think that this information should be put in the conclusion as a potential next step in optimization of bioprocess.

Lines 105-107: I think that the main purpose is to check if whey permeate will be appropriate feedstock for SA production, but not improving the SA productivity. Am I right?

Materials and Methods

Lines 120-130: The description should be supplemented by culture temperature. Please provide also a condition of culture (aerobic/anaerobic) as well as the composition of the medium used and the company.

Line 137: "...were called A,B,C" It should be A,B,C, respectively. 

Lines 143-144: "Lactose concentrate, yeast extract and fermentation media were chemically characterized prior to their use." This sentence sounds like there were used lactose concentrate, yeast extract and fermentation media. Meanwhile, while I am reading this paper I think the fermentation medium was composed by two ingredients: whey concentrate and YE? Moreover, how were these ingredients chemically characterized?

Lines 147-148: Why were these temperature and pH value of the medium used? Citation?

Lines 161-162: Did you detect some amount of ethanol?

Lines 166-169: Please, complete the description with additional information, such as column thermostat temperatures, running phase, speed (0.5 ml/min?).

Line 171: Why did you use the standard of fructose and arabinose?

Lines 172-174: What do you mean by pretreatment of lactose, YE and fermentation media and for what did you use cold water extraction? To prepare a sample to HPLC analysis, the syringe filter can just be used.

Line 173: the bioreactor that was used was not equipped with a pH electrode?

Lines 194: Why did you provide the amount of sugars regarded as glucose equivalents?

Results

In general, I believe that all tests should be carried out on pure lactose, and the use of whey concentrate should be the final (additional) stage of the research. Because by using a different concentrate, we can get different results.

Table 1: In 1 kg there is only 268 g of sugars. What is the rest? Generally, the whey permeate usually contains about 70-80% lactose.

Lines 254-267: I think that there is need to compare the results obtained by authors with other results in the literature but with that where the autors were using lactose, not other waste feedstock. I believe that the authors should add a table in which you can compare the results obtained with the results for other bacterial producers using lactose, for example:

Szczerba, H., Komoń-Janczara, E., Dudziak, K. et al. A novel biocatalyst, Enterobacter aerogenes LU2, for efficient production of succinic acid using whey permeate as a cost-effective carbon source. Biotechnol Biofuels 13, 96 (2020). https://doi.org/10.1186/s13068-020-01739-3

Lines 322-324: This is contrary to the model described in the introduction.

Author Response

We like to thank you for your valuable comments and hints. Please find in the following the particular answers to your remarks. Thank you for helping us to improve our manuscript.

Abstract

Lines 18-19: ".... lactose concentrate, ....". Do you think about whey permeate?

We agree that the term lactose concentrate was dot explained clear enough before. A whey permeate was used, which was concentrated by evaporation by the company Karwendel-Werke-Huber GmbH. We have explained this now in the materials and methods section as follows:  

Page 3, Line 140: Lactose concentrate, a concentrate from whey permeate and a by-product from cheese-making (Karwendel-Werke-Huber GmbH & Co. KG, Buchloe, Germany), and yeast extract (Ohly KAT, Deutsche Hefewerke GmbH, Nürnberg, Germany) were used for the preparation of different fermentation media. The lactose concentrate is currently used as a substrate in biogas plants. To save transport costs, it was concentrated by the Karwendel-Werke-Huber GmbH to a dry matter content of about 30 %.

Introduction

Lines 36: Succinic acid is very rarely mentioned as VFA.

Page 1, Line 36: Instead of “volatile fatty acid” we now use the term “compound”.

Lines 41-43: Please provide more up-to-date data: 

We actualized the information on the global market size of bio-based succinic acid given in the manuscript and rephrased the respective sentence as follows:

Page 1, Line 43: In 2020, the annual global production of bio-based succinic acid was valued at 215 million USD and it is expected to grow with a compound annual growth rate of 11 % between 2021 and 2027 [4].

Lines 47-50: Did you perform experiments using municipal solid wastes, sugarcane bagasse, carob pods and so on in this paper? Because that's what it sounds like.

The respective sentence refers to the work of others, who applied the mentioned substrates. In the manuscript, the respective sentence was started with “In recent studies…” and the respective references were given as well. To make it less confusing, the sentence has been rephrased in the present manuscript as follows:

Page 2, Line 49: “In recent publications scientists presented research on succinic acid production from the organic fraction of municipal solid waste [7], sugarcane bagasse [8], carob pods [9], corn stover [10], and straw [11], using the succinogenic bacterial strains A. succinogenes or B. succiniciproducens.

Lines 89-94: I think that this fragment is not necessary cause the present study is regarding to lactose.

This information was important to us in respect to the supplementation of the fermentation medium with yeast. To emphasize this aspect, the following sentence has been added in the present manuscript:

Page 2, Line 98: “Although the present study was not focused on glucose, but on lactose, the aforementioned information indicates that the supplementation with yeast might improve a lactate based medium too.

Lines 112-116: I think that whey permeate is a rich source of amino acids and trace metals. Hence, I believe that the use of yeast extract does not have to be explained by the fact that the whey permeate is poor. I believe also that the use of YE is fully justified in the preliminary research and the authors do not have to justify that its limitation is needed.

We agree with the reviewer and have removed the respective sentence. Since neither proteins nor trace metals represent a deficiency, we now use the expression "essential nutrients". This was corrected accordingly in several places in the manuscript.

Lines 95-100: This information is no needed, as there is no experiments on elimination of side products in this paper. I think that this information should be put in the conclusion as a potential next step in optimization of bioprocess.

The respective sentence has been removed from the present manuscript.

Lines 105-107: I think that the main purpose is to check if whey permeate will be appropriate feedstock for SA production, but not improving the SA productivity. Am I right?

We agree with the reviewer and have rephrased the respective sentence in the present manuscript as follows:

Page 3, Line 111: “The main objective of the present study was to investigate the suitability of lactose concentrate as feedstock for the two natural acid producers A. succinogenes 130Z (DSM 22257) and B. succiniciproducens (DSM 22022). Doing this, fermentation tests were performed in batch mode using lactose concentrate, generated from cheese whey, as feedstock and main compound of the fermentation media.

Materials and Methods

Lines 120-130: The description should be supplemented by culture temperature. Please provide also a condition of culture (aerobic/anaerobic) as well as the composition of the medium used and the company.

We agree with you and have rephrased the sentence as follows:

Page 3, Line 135: “Precultures were produced by inoculation of 100 mL tryptic soy broth (Merck KGaA, Darmstadt, Germany) in shaker flasks within 24 hours at 37°C under aerobic conditions.

Line 137: "...were called A,B,C" It should be A,B,C, respectively. 

We corrected it in the present manuscript.

Page 3, Line 145: “For the preparation of media, the lactose concentrate was diluted with water to adjust three different target sugar concentrations of 20, 40 and 60 ± 5 g L-1, the resulting media were called A, B and C respectively.

Lines 143-144: "Lactose concentrate, yeast extract and fermentation media were chemically characterized prior to their use." This sentence sounds like there were used lactose concentrate, yeast extract and fermentation media. Meanwhile, while I am reading this paper I think the fermentation medium was composed by two ingredients: whey concentrate and YE? Moreover, how were these ingredients chemically characterized?

We thank you for this comment. The medium contained lactose concentrate and yeast extract. We deleted the particular sentence in the present manuscript.

Lines 147-148: Why were these temperature and pH value of the medium used? Citation?

We completed the manuscript as follows:

Page 4, Line 158: “As optimal growth conditions, the literature states for both strains mesophilic temperatures (37°C) [15,16] and pH values of 6.8 [9,22] or 6.5 [14]. Based on this information, a temperature of 37°C and a pH value of 6.7 was applied.

Lines 161-162: Did you detect some amount of ethanol?

The concentration of ethanol was not determined. The HPLC system, used for the determination of the different acids produced during fermentation, was not equipped for the analysis of alcohols.

Lines 166-169: Please, complete the description with additional information, such as column thermostat temperatures, running phase, speed (0.5 ml/min?).

We completed the present manuscript as follows:

Page 4, Line 184: The temperature of the column thermostat was 40°C; the eluent, 0.01 N sulfuric acid, was pumped through the column with a speed of 0.8 mL min-1.

Line 171: Why did you use the standard of fructose and arabinose?

We agree with you that both sugars are not relevant for the present study. Therefore, we have removed the respective sentence.

Lines 172-174: What do you mean by pretreatment of lactose, YE and fermentation media and for what did you use cold water extraction? To prepare a sample to HPLC analysis, the syringe filter can just be used.

We agree with you and have rephrased the respective sentence as follows:

Page 4, Line 191: “Before samples of lactose concentrate and fermentation medium were analyzed by HPLC, they were diluted with water. The dilution factor of the samples was determined by the present concentration of the analyzed sugars (glucose, galactose, saccharose and lactose) and acids (succinic, acetic, formic and lactic acid). The achieved concentration of every sugar and acid component had to fit the measuring range between 0.5 and 5.0 g L-1. The HPLC analysis of yeast extract required a pretreatment by protein precipitation with Carrez reagent as described by [28].

Line 173: the bioreactor that was used was not equipped with a pH electrode?

We completed the present manuscript with information about the equipped pH electrode.

Page 4, Line 161: The pH value was measured with the sensor EasyFerm Plus PHI K8 225 (Hamilton Bonaduz AG, Bonaduz, Switzerland) and automatically regulated by the addition of 5 N NaOH.  

Lines 194: Why did you provide the amount of sugars regarded as glucose equivalents?

The declaration of sugars regarded as glucose equivalents seemed to us beneficial because this unit enables the calculation of the sugar consumption from the obtained data.

Results

In general, I believe that all tests should be carried out on pure lactose, and the use of whey concentrate should be the final (additional) stage of the research. Because by using a different concentrate, we can get different results.

Your suggestion is comprehensible. As responded to Reviewer 2, we would like to point out again that there are already publications that demonstrate the use of lactose-rich substrates (whey). The works given are a good basis for comparison in order to determine the experimental framework. For example: We discussed some articles related to yeast supplementation in the introduction. We used this as a guide to determine the amount of yeast to be added.

Another example relates to E. aerogenes. A similar fermentation velocity is described for this E. aerogenes as for the strains described by us, which increases the confidence in the results presented here. We have highlighted this now in more detail:

Page 9, line 365: “With up to 140 g L-1, the sugar concentration applied by [36] was more than twice as high as in the experiments presented here. Moreover, the fermentation with E. aerogenes was much slower, needing approximately 144 h to terminate. The here presented fermentation experiments with A. succinogenes and B. succiniciproducens finished in less than 50 h. To reach a similar velocity with E. aerogenes, the authors of the respective study had to lower the lactose concentration to 60 g L-1, which is in a similar range as the here presented results [36].

Finally, we would like to mention that the results of the conducted extensive chemical analysis showed that the lactose contained in the media was actually converted.

Table 1: In 1 kg there is only 268 g of sugars. What is the rest? Generally, the whey permeate usually contains about 70-80% lactose.

The dry matter content of the lactose concentrate was about 30 %. The rest of the fresh mass was water. According to an employee of the Karwendel-Werke-Huber GmbH, the lactose concentrate was used as feedstock for anaerobic digestion. The whey permeate was concentrated to reduce the transportation costs. This is now described in the manuscript as follows:

Page 3, line 140: “Lactose concentrate, a concentrate from whey permeate and a by-product from cheese-making (Karwendel-Werke-Huber GmbH & Co. KG, Buchloe, Germany), and yeast extract (Ohly KAT, Deutsche Hefewerke GmbH, Nürnberg, Germany) were taken for the preparation of different fermentation media. The lactose concentrate is currently used as a substrate in biogas plants. To save transport costs, it was concentrated to a dry matter content of about 30 %.

Lines 254-267: I think that there is need to compare the results obtained by authors with other results in the literature but with that where the autors were using lactose, not other waste feedstock. I believe that the authors should add a table in which you can compare the results obtained with the results for other bacterial producers using lactose, for example:

Szczerba, H., Komoń-Janczara, E., Dudziak, K. et al. A novel biocatalyst, Enterobacter aerogenes LU2, for efficient production of succinic acid using whey permeate as a cost-effective carbon source. Biotechnol Biofuels 13, 96 (2020). https://doi.org/10.1186/s13068-020-01739-3

We agree with this comment of you. Reviewer 2 also emphasized that the same publication should be taken into account. The corresponding reference is now given and has been compared with the generated results.

We have now discussed the reference suggested by the reviewers 2 and 3 in the manuscript as follows:

Page 9, Line 353: “Comparing the results of the present study obtained with A. succinogenes and B. succiniciproducens to other research articles on succinic acid fermentation, it becomes clear that there are multiple other strains, which can be used for this purpose. For example, it was shown that Mannheimia succiniciproducens, a strain that is genetically close to Basfia succiniciproducens, is also able the ferment lactose efficiently [34]. The fact that these two strains showed a similar fermentation performance increases the trustworthiness and shows reproducibility [35]. Another strain, which has been described in literature as particularly suitable is Enterobacter aerogenes LU2 [36]. The authors of the mentioned publication have shown that E. aerogenes is also able to utilize lactose, in their experiments succinic acid concentrations up to 57.7 g L-1 were achieved. With succinic acid yields between 12 and 24 g L-1, much lower concentrations were achieved with A. succinogenes and B. succiniciproducens in the present study. However, it must be considered that the sugar content of the growth medium differed clearly. With up to 140 g L-1, the sugar concentration applied by [36] was more than twice as high as in the experiments presented here. Moreover, the fermentation with E. aerogenes was much slower, needing approximately 144 h to terminate. The here presented fermentation experiments with A. succinogenes and B. succiniciproducens finished in less than 50 h. To reach a similar velocity with E. aerogenes, the authors of the respective study had to lower the lactose concentration to  60 g L-1, which is in a similar range as the here presented results [36].

Lines 322-324: This is contrary to the model described in the introduction.

We agree with you that the respective phrase was not clearly expressed. We have explained the supplementation with yeast extract more clearly now:

Page 8, Line 341: “The results indicate a lack of essential nutrients that are contained in yeast extract. [33] investigated the supplementation of the medium with yeast extract during fermentation of high-sugar-content beverages with A. succinogenes. The concentration of yeast extract was 5 g L-1, the initial sugar concentration varied between 46 and 56 g L-1. The omission of yeast extract resulted in a decrease of the fermentation performance depending on the medium (pineapple juice or ACE juice) by 14 % or 13 % (final concentration of acid mixture), 21 % or 16 % (sugar consumption) and 3 % or 14 % (succinic acid concentration), respectively. Therefore, the importance of yeast extract for fermentation of sugars is clearly recognizable in [33], this supports the results of the present study.